# Prevalence and correlates of compliance with 24-h movement guidelines among children from urban and rural Kenya—The Kenya-LINX project

Nils Swindell[1]*, Lucy-Joy Wachira[2], Victor Okoth[2], Stanley Kagunda[2], George Owino[3], Sophie Ochola[4], Sinead Brophy[5], Huw Summers[6], Amie Richards[1], Stuart J. Fairclough[7], Vincent Onywera[2], Gareth Stratton[1]

1 Applied Sports Technology Exercise and Medicine Research Centre, Swansea University, Swansea, Wales, United Kingdom, 2 Department of Physical Education, Exercise and Sports Science, Kenyatta University, Nairobi, Kenya, 3 Department of Sociology, Gender and Development Studies, Kenyatta University, Nairobi, Kenya, 4 Department of Food, Nutrition and Dietetics, Kenyatta University, Nairobi, Kenya, 5 National Centre for Population Health and Wellbeing Research, Swansea University Medical School, Swansea, Wales, United Kingdom, 6 Department of Biomedical Engineering, Swansea University, Swansea, Wales, United Kingdom, 7 Department of Sport and Physical Activity, Movement Behaviours, Nutrition, Health, & Wellbeing Research Group, Edge Hill University, Ormskirk, United Kingdom

* n.j.swindell@swansea.ac.uk

**Data Availability Statement:** Ethical approval for this study was granted by the Swansea University College of Engineering Research Ethics Committee

## Abstract

### Background

Like many countries in sub-Saharan Africa, Kenya has experienced rapid urbanization in recent years. Despite the distinct socioeconomic and environmental differences, few studies have examined the adherence to movement guidelines in urban and rural areas. This cross-sectional study aimed at examining compliance to the 24-hour movement guidelines and their correlates among children from urban and rural Kenya.

### Method

Children (n = 539) aged 11.1 ± 0.8 years (52% female) were recruited from 8 urban and 8 rural private and public schools in Kenya. Physical activity (PA) and sleep duration were estimated using 24-h raw data from wrist-worn accelerometers. Screen time (ST) and potential correlates were self- reported. Multi-level logistic regression was applied to identify correlates of adherence to combined and individual movement guidelines.

### Results

Compliance with the combined movement guidelines was low overall (7%), and higher among rural (10%) than urban (5%) children. Seventy-six percent of rural children met the individual PA guidelines compared to 60% urban children while more rural children also met sleep guidelines (27% vs 14%). The odds of meeting the combined movement guidelines reduced with age (OR = 0.55, 95% CI = 0.35–0.87, p = 0.01), was greater among those who could swim (OR = 3.27, 95% CI = 1.09–9.83, p = 0.04), and among those who did not

and The Kenyatta University Ethical Review Committee, provided that participants' data was only accessible by the research team. Data from this research study contains information at the individual and school level which when combined could be used to identify individuals. Participants of this study did not consent to having their data publicly available. Requests to access the data may be directed to the Swansea University College of Engineering Research Ethics Committee coe-researchethics@swansea.ac.uk and the Ethics and scientific review committee at Kenyatta University chairman.kuerc@ku.ac.ke, APPLICATION NUMBER PKU2106/11254.

**Funding:** This research was funded by the British Academy under the urban infrastructure of wellbeing scheme https://www.thebritishacademy.ac.uk/programmes/urban-infrastructures-well-being Grant number: UWB190069 The grant was awarded to Gareth Stratton (PI), Vincent Onywera, George Owino, Lucy-Joy Wachira, Huw Summers and Sinead Brophy. The funders had no role in study design, data collection and analysis, decision to publish, or preparation of the manuscript.

**Competing interests:** The authors have declared that no competing interests exist.

engage in ST before school (OR = 4.40, 95% CI = 1.81–10.68, p<0.01). The odds of meeting PA guidelines increased with the number of weekly physical education sessions provided at school (OR = 2.1, 95% CI = 1.36–3.21, p<0.01) and was greater among children who spent their lunch break walking (OR = 2.52, 95% CI = 1.15–5.55, p = 0.02) or running relative to those who spent it sitting (OR = 2.33, 95% CI = 1.27–4.27, p = 0.01).

## Conclusions

Prevalence of meeting movement guidelines among Kenyan children is low and of greatest concern in urban areas. Several correlates were identified, particularly influential were features of the school day, School is thus a significant setting to promote a healthy balance between sleep, sedentary time, and PA.

## Introduction

Sufficient PA and sleep, combined with limited sedentary behaviour is important for children's physical, cognitive and social-emotional development [1]. For example, movement behaviours (e.g., sleep sedentary time and PA) have been collectively associating with motor skills (physical), memory and vocabulary development (cognitive), conduct problems, peer problems and prosocial behaviour (social-emotional) in children [1, 2]. In adolescents, adherence to the 24-hour movement guidelines and especially recommendations for PA and sleep duration contributes positively to increased academic performance [3].

In contrast, a lack of PA, high sedentary behaviour, and insufficient sleep are adversely associated with adiposity, cardiometabolic health, fitness, as well as mental, social, and emotional wellbeing in children and adolescents [4]. Furthermore, physical inactivity contributes to the development of many of the world's non-communicable diseases (NCDs), including type 2 diabetes and cardiovascular disease and represents a major public health challenge [5].

The daily movement behaviours PA, sedentary behaviour, and sleep comprise parts of a finite whole (24 hours) and are therefore interconnected and should not be considered in isolation. There is an increasing recognition of the interrelationship between all three behaviours and their combined impact on health outcomes at all ages [4]. In line with this concept, there is now a shift towards 24-hour movement guidelines which have been adopted for children in six countries and by the World Health Organization [4]. Guidelines recommend that children and adolescents spend at least 60 minutes in moderate-to-vigorous PA (MVPA), accumulate no more than 2 h of recreational ST, and get sufficient sleep (9–11 h for children and 8–10 h for adolescents) per day [6].

Despite compelling evidence for the importance of 24-hour movement behaviour guidelines for physical, mental, and social health [7], the prevalence of meeting these guidelines remains low [8–11]. While studies from sub-Saharan Africa (SSA) and developing countries in general that utilise device-based movement behaviour measures are scarce, the Kenyan sample of the International Study of Childhood Obesity, Lifestyle and Environment (ISCOLE), were the most physically active of the 12 countries in the ISCOLE study. Yet, only 6.5% of children aged 9–11 years met recommendations for PA, sedentary time, and sleep, while 16.3% met none [9]. The ISCOLE study sample was selected from urban areas (Nairobi) and may not be representative of the broader population of which (72%) live in rural areas [12].

In Kenya, like other countries in SSA, economic and demographic changes are leading to a decline in PA and increased sedentary behaviour [13, 14]. The 2016 Active Healthy Kids (AHK) -Kenya Report card revealed that only half of Kenyan children and youth met the global guidelines for PA [15]. Furthermore, children in urban Kenya were less physically active and spent more time in recreational ST than their rural counterparts [13, 15] Worryingly, the prevalence of childhood overweight/obesity has increased in parallel with the behavioural changes that have accompanied rapid urbanization [16].

Deeper insight into movement behaviours including in urban and rural areas is needed to better understand the impacts of urbanization and inform local policies and strategies that address the PA transition.

Children and adolescents spend most of their waking day at school, therefore, schools are uniquely positioned to promote a healthy active lifestyle to children of different ages and socio-economic backgrounds [17]. Through sports participation, play and PE, schools offer many opportunities for children to be physically active and develop skills that support a healthy active future. In western countries, physical education (PE) and participation in school sports are associated with total PA [18]. However, little is known about how school policy and practice impact on compliance with movement guidelines in SSA.

Kenya's National Physical Activity Plan [19] and School Health Policy [20] advocate for the promotion of PA at school through the provision of PE, integrating PA into school policy and offering students the opportunity to engage in various types of PA. School intervention efforts require an understanding of the contextual factors at the individual and school level associated with adherence to 24h movement behaviour guidelines. Therefore, this will be the first study to i) examine the prevalence of adherence to the combined 24-h movement guidelines among children from urban and rural Kenya, and ii) identify the individual and school level correlates of meeting these guidelines.

## Materials and methods

Nairobi City County, as the largest and capital city of Kenya was selected to represent the urban location while Kitui County, situated 170 kilometres east of Nairobi, and where 95% of the population reside rurally was the rural location [21]. Year 5 and 6 primary school children (Kitui, n = 207, 52.7% female, 11.3 ± 0.9 years; Nairobi n = 287, 51.2% female, 10.9 ± 0.8 years) were recruited from 8 urban (Nairobi City County) and 8 rural (Kitui County) non-boarding schools. Five private (fee paying) and three public (free to attend) primary schools were selected from across three sub-counties of the respective counties to ensure a wider distribution of the sampled schools. Stratified, random cluster sampling was used to ensure appropriate regional representation of public and private schools for each county and ensure dispersion of schools across sub counties. A Full class participation approach was used including all children of eligible age who returned signed parental consent.

The study was approved by the Kenyatta University and Swansea University Ethical Review Committees [Reference number: PKU2106/11254], research permits, and authorization were issued by the National Commission for Science, Technology, and Innovation (Kenya) [License No: NACOSTI/P/20/5030]. Further clearance was obtained from County and Sub County Education offices in Nairobi (Ref: GL/NC/142.VOL VI/345) and Kitui (Ref. No: KTIC/ED/Res/Vol. l/22/113). Informed parental consent and child assent was attained before research commenced.

Stature (to the nearest 0.01m) and body mass (to the nearest 0.1 kg) were assessed using a stadiometer (SECA 217 Hamburg, German) and weighing scale (SECA 813 electronic flat scale, Hamburg, German) respectively, while the participant was lightly dressed and barefoot.

BMI was calculated by dividing body weight with the squared height in meters (kg/m2). Standardised BMI scores were calculated using the International Obesity Task Force values [22] underweight, healthy weight, overweight, and obesity were defined according to International Obesity Task Force criteria.

## Physical activity and sleep

Participants were asked to wear an Axivity AX3 (Axivity Ltd, Newcastle, UK) accelerometer on the non-dominant wrist for 24 h·day$^{-1}$ for 7 days with accelerations recorded at 100 Hz and a dynamic range of +/- 8 g. Data were downloaded using OmGui open-source software (OmGui Version 1.0.0.43, Open Movement, Newcastle University, UK). All data were processed in R (http://cran.r-project.org) using GGIR v2.3.0 [23]. Signal processing included auto-calibration using local gravity as a reference [24] and the detection of non-wear. To converted raw triaxial accelerations into one omnidirectional measure of acceleration the Euclidean Norm minus 1g, (ENMO) was calculated from raw accelerations from the three axes minus 1$g$ which represents the value of gravity (i.e., ENMO = $\sqrt{(x^2 + y^2 + z^2)}$– 1). ENMO values were averaged over 5-seconds epochs for further analysis [25].

Accelerometer non wear time periods were determined based on the SD and value range of the accelerations at each axis, calculated for 60-minute windows with a 15-minute sliding window [25]. If for 2 out of the 3 axes the SD was less than 13 mg or the value range was less than 50 mg, the time window was classified as non-wear. Non wear data was imputed by the average at similar time points on other days of the week. Participants were excluded if less than four valid days of wear (i.e., at least 16 h·day$^{-1}$) were recorded [26] and/or if accelerometer post-calibration error was $>$ 10 m$g$ [24].

Child specific cut-points based on ENMO values were applied to define time accumulated above an acceleration of $>$200 mg [27] to estimate time spent in MVPA ($>$3 METS). Sleep duration was estimated using a polysomnography-validated algorithm based on the distribution of change in the z-angle [28].

## Screen time

Children completed a self-report online survey during their school day while under supervision of their teachers and research staff. The survey was adapted from the Child Health and Activity Tool (CHAT) [29] for use in Kenya. CHAT has been used extensively in this age group and has shown acceptable validity [30]. To capture their typical screen use, participants were asked the following questions:

*"On how many days did you watch TV for two or more hours per day?" and "On how many days did you play video or computer games or mobile phone games or use a computer for something that is not schoolwork for two or more hours per day?"* Questions were separated for week and weekend days and the total number of days on which participants exceeded 2-hours of ST was calculated.

Participants adhered to the combined movement guidelines if they accumulate at least 60 min of MVPA, did not exceed 2 hours of recreational ST and recorded 9–11 hours of sleep on each day of the week.

## Correlates

Potential correlates were derived from the CHAT survey in which children are required to recall their previous day's behaviour, through a timeline of their typical school day. To capture how children spend their time before school, the following questions were used: *Before the lessons started yesterday how long did you spend doing sports or exercise? Before the lessons started*

*yesterday, how long did you spend sitting down watching TV/using mobile phone /tablet or internet? Before the lessons started yesterday, how long did you spend doing homework/reading?* The responses were as follows: *10, 20, 30, 40, 50, 60, >60 minutes*. Responses were dichotomised to indicate if children spent any time (yes/no) in the above behaviours.

Children were asked how they travelled to and from school: *How did you get to school yesterday morning? How did you get home yesterday?* The following responses *I walked, I ran, and I rode a bicycle* were combined to indicate if children used an active mode of transport.

To capture how children spend their time during school, the following questions were used: *What did you do for most of your morning break time yesterday? Apart from eating your lunch, what did you do for MOST of your Lunch break yesterday?* Responses included: *I sat around, I stood around, I walked around, I ran around.*

To capture how children spend their time after school, the following questions were included: *After school yesterday, how long did you spend doing any physical activity, sports or exercise? After school yesterday, how long did you spend sitting down watching TV/using mobile phone/tablet or internet? After school yesterday, how long did you spend doing homework/reading?* The responses were as follows: *10, 20, 30, 40, 50, 60, >60 minutes*. Responses were dichotomised to indicate if children spent any time (yes/no) in the above behaviours.

Further, children were asked if they were engaged in any sport clubs or groups both in and outside of school two questions on their physical competence.

*Are you a member of school clubs, Brownies, Guides or Scout? What school sport clubs of physical activities do you take part in at least once a week? In school (this does not include P. E.) What school sport clubs of physical activities do you take part in at least once a week? Out of school.*

*Can you ride a bicycle?*

*Can you swim?*

One teacher from each participating school completed a questionnaire on the school policies and practices which included the following questions:

*Does your school have written policy/regulations/directive/practices, concerning physical activity?*

*Does your school have a committee that oversees or offers guidance on the development of guidelines/regulations and practices concerning physical activity and healthy eating at your school?*

*Does your school offer school bus transportation to pupils who participate in extracurricular activities?*

*How many breaks of 15 to 29 minutes do pupils in class/grade 4/5 have in a day?*

Available responses: *0,1,2, 3 or more.*

*How many breaks of 30 minutes or more do pupils in class/grade 4/5 have in a day?* Available responses: *0,1,2, 3 or more*

*How many physical education sessions do pupils participate in per week? How long are PE sessions?*

## Statistical analysis

All analyses were performed using R (version 4.0.2) and R Studio (version 2022.02.0+443).

Descriptive characteristics of participants were summarized using means (SD) or frequencies. Potential differences between participants from urban and rural areas were examined using unpaired t-tests for continuous variables and chi-square tests ($\chi^2$) for categorical variables using the $\chi^2$ test function. Significant associations (p<0.05) were followed by pairwise comparisons using the $\chi^2$.posthoc.test function from the $\chi^2$.posthoc.test package. Bonferroni-adjusted *P* values were used to account for multiple comparisons.

Univariate binary logistic regression was conducted to examine associations between potential correlates and adherence to individual and combined movement guidelines. Any explanatory variable with p<0.1 was retained for multivariate analysis.

Multi-level multivariate logistic regression analysis was performed with the 'lme4' package (v. 1.1–25) to identify correlates of adherence to the individual and combined movement guidelines. To account for the clustered nature of the data (children within schools), potential correlates were included as fixed effects while school was included as a random effect. Odds ratios and the 95% confidence intervals corresponding to the individual correlates as well as their significance were calculated.

A backward stepwise approach was used, whereby the correlate with the highest *p*-value was sequentially removed from the model until all correlates had a *p*-value below 0.10. To quantify the severity of multicollinearity in the model, generalized variance inflation factors (GVIF) were computed using the "VIF" function in R. The GVIF values suggested high collinearity between school level variables. Where two or more variable were highly collinear, each variable was tested in the model independently those with best fit and satisfied Akaike's information criterion (AIC) were retained in the model.

## Results

Descriptive characteristics of the participants are shown in Table 1 stratified by urban and rural areas. Children from urban schools were significantly younger (<0.01) and had a greater age adjusted BMI (<0.01) than children from rural schools. Significant differences between the proportion of urban and rural children meeting movement guidelines were observed for PA and sleep. Urban children were less likely to meet both guidelines (p < 0.01 for both).

The proportions of participants not meeting guidelines, and meeting the MVPA, ST, and sleep duration recommendations, as well as various combinations of 24h recommendations are presented in Fig 1.

Potential correlates can be found in the supplementary material (S1 Table) stratified by urban and rural areas. Significant differences were found in the way children from urban and rural schools spent their time before and during school. Specifically, more children from urban schools reported ST and doing homework before school than children from rural school

**Table 1. Descriptive characteristics.**

|  | Urban (n = 287) | Rural (n = 193) | t-value/chi-squared | P value |
|---|---|---|---|---|
| **Age (years)** | 10.9 (0.74) | 11.3 (0.95) |  | <0.01 |
| **Sex (% female)** | 52.4 | 54.0 | 0.17 | 0.68 |
| **BMI-z** | 0.23 (1.0) | -0.96 (1.3) | -9.14 | <0.01 |
| **MVPA (% meeting recommendation)** | 60.3 | 75.6 | 12.3 | <0.01 |
| **Sleep (% meeting recommendation)** | 14.6 | 27.5 | 11.96 | <0.01 |
| **Screen (% meeting recommendation)** | 56.0 | 60.6 | 1.00 | 0.32 |

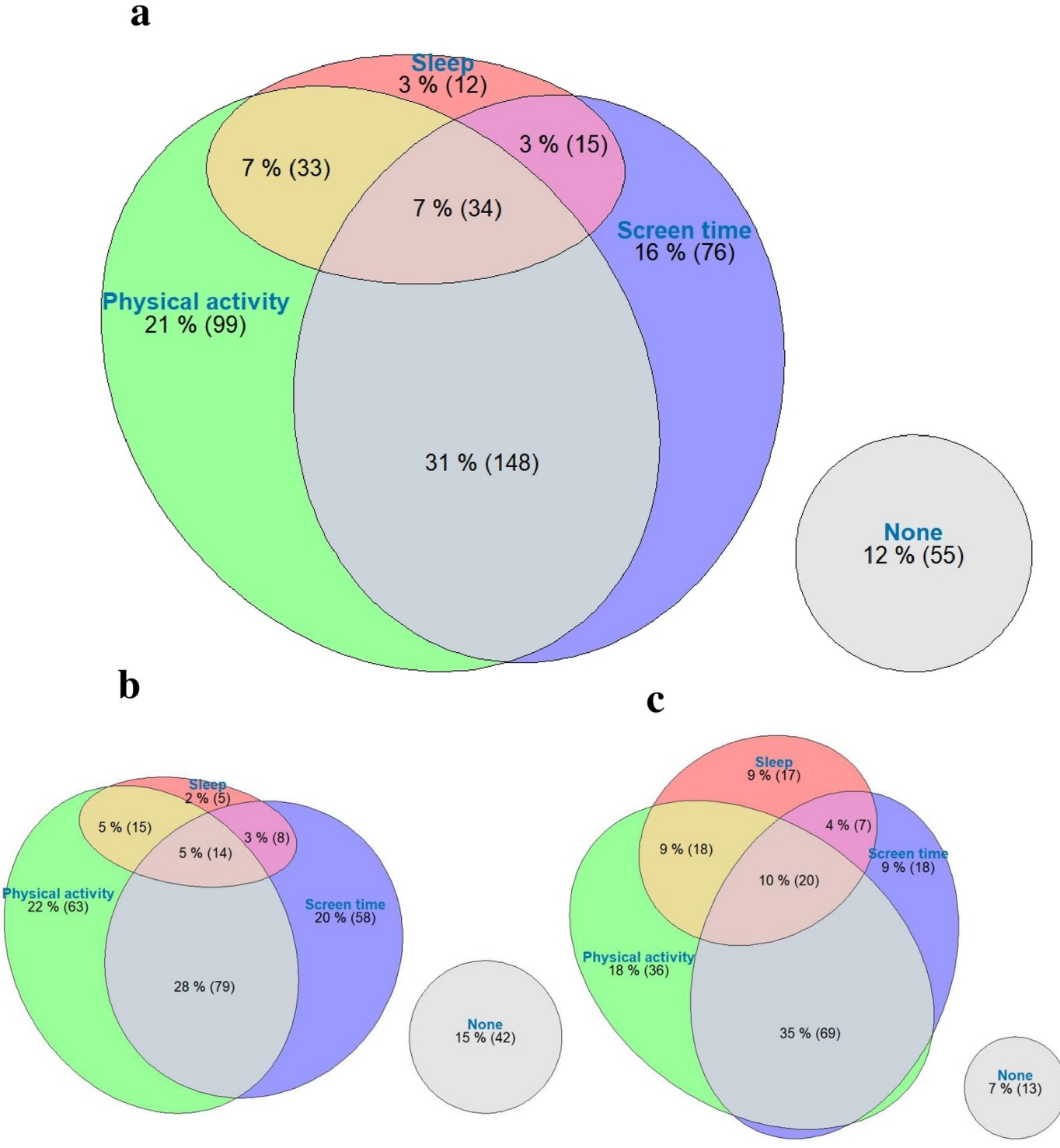

**Fig 1. Venn diagram showing the percentage of children (N = 481) meeting the physical activity screen time and sleep recommendations and the combination of these behaviours.** The ellipse size is proportional to the proportion meeting the recommendations. **a** all participants, **b** rural participants, **c** urban participants.

and BMI; Body Mass Index, MVPA; moderate-to-vigorous physical activity significantly more children from urban schools spent their school break sitting while fewer spent it running (all p<0.01). The same was true for the lunch break with significantly fewer children from urban schools spending their lunch break running (p = 0.05). After school, significantly more children from urban schools reported ST and doing homework than their rural counterparts (p = 0.03 and <0.01). However, children from urban schools were more likely to ride a bike (p = 0.02) and attend Brownies, Guides or Scouts (p = 0.03). There was no difference in the prevalence of attending sports clubs either inside or outside school (p = 0.93).

Significant differences were also found at the school level between urban and rural schools. The school day in urban schools was significantly longer with fewer breaks, fewer PE sessions per week and PE session were more likely to be of shorter duration than in rural schools. However, urban schools were more likely to offer transport for extracurricular activities and have a school health committee that oversees the promotion of diet and PA (all p<0.01).

The univariate analysis of associations between the potential correlates and adhering to the combined movement guidelines, meeting any two recommendations or each individual guideline can be found in the supplementary material (S2 Table). A total of 24, 19, and 12 potential correlates reached a p-value of less than 0.1 for meeting at least one, a combination of two, or all movement guidelines respective and were retained for further analysis.

The results of the multi-level multivariable analysis are shown in Table 2. The odds of meeting the combined movement guidelines decreased significantly with age (OR = 0.55, 95% CI = 0.35–0.87, p = 0.01), and were 4 times greater among children who did not engage in any ST before school (OR = 4.40, 95% CI = 1.81–10.68, p<0.01). Further, children who could swim had significantly higher odds of meeting the combined movement guidelines than those who could not (OR = 3.27, 95% CI = 1.09–9.83, p = 0.04).

Children from Nairobi County had significantly lower odds of adherence to the combined PA and sleep guidelines (OR = 0.27, 95% CI = 0.07–0.95, p = 0.04) after controlling for differences in age and weight status. Boys had significantly greater odds of meeting PA guidelines (OR = 3.82, 95% CI = 2.21–6.63, p<0.01) and a combination of PA and screen guidelines (OR = 1.64, 95% CI = 1.10–2.47, p = 0.02) than girls. Children who were above a healthy weight for their age and sex had significantly greater odds of meeting the PA guidelines OR = 0.39, 95% CI = 0.18–0.84, p = 0.02). On the other hand, children who were below a healthy weight were also less likely to meet the combination of PA and sleep guidelines (OR = 0.33, 95% CI = 0.11–0.97, p = 0.04).

The odds of meeting PA guidelines was significantly associated with school activities. Specifically, the odds of meeting PA guidelines increased by 110% for every additional PE session provided per week (OR = 2.1, 95% CI = 1.36–3.31, p<0.01). Furthermore, children who spent most of their school lunch break walking around or running around had significantly greater odds of meeting the guidelines relative to children who spent their school lunch break sat around (OR = 2.52, 95% CI = 1.15–5.55, p = 0.02 and OR = 2.33, 95% CI = 1.27–4.27, p = 0.01, respectively). The same was true for meeting the combined PA and ST guidelines and even spending the lunch break standing rather than sitting was beneficial (OR = 2.76, 95% CI = 1.06–7.15, p = 0.04).

The provision of transport for extracurricular activities was associated with reduced odds of meeting PA guidelines (OR = 0.31, 95% CI = 0.13–0.76, p = 0.01) while active transport to school and being member of a school sports club were associated with greater odds of achieving the screen guidelines (OR = 1.51, 95% CI = 1.02–2.06, p = 0.04 and OR = 2.64, 95% CI = 1.45–4.03, p<0.01 respectively). The odds of achieving the sleep guidelines increased with the number of PE sessions provided per week (OR = 1.57, 95% CI = 1.16–2.11, p = 0.01). Also, those with more breaks at school were less likely to achieve sleep guidelines (OR = 0.32, 95%

**Table 2. Multi-level multivariable analysis of correlates of meeting 24-h movement guidelines.**

| | %[a] | OR | 95% CI | | p |
|---|---|---|---|---|---|
| PA (N = 378) | | | | | |
| **Sex (female)** | 54 | | | | |
| *Male* | 80 | 3.82 | 2.21 | 6.63 | < .01 |
| **Weight status (Healthy weight)** | 69 | | | | |
| *Below a healthy weight* | 76 | 0.9 | 0.50 | 1.81 | 0.88 |
| *Above a healthy weight* | 42 | 0.39 | 0.18 | 0.84 | 0.02 |
| **Lunch break (Sat around)** | 57 | | | | |
| *Ran around* | 75 | 2.33 | 1.27 | 4.27 | 0.01 |
| *Stood around* | 67 | 2.02 | 0.55 | 7.34 | 0.29 |
| *Walked around* | 63 | 2.52 | 1.15 | 5.55 | 0.02 |
| **School provision of transportation for extracurricular activities (no)** | 75 | | | | |
| *Yes* | 59 | 0.31 | 0.13 | 0.76 | 0.01 |
| **PE Sessions per week** | | 2.1 | 1.36 | 3.21 | < .01 |
| Screen (N = 472) | | | | | |
| **Screen time before school (yes)** | 49 | | | | |
| *No* | 66 | 1.49 | 1.01 | 2.26 | 0.04 |
| **Active transport to school (yes)** | 62 | 1.51 | 1.02 | 2.26 | 0.04 |
| *No* | 51 | | | | |
| **Screen time after school (no)** | 79 | 3.24 | 2.07 | 5.45 | < .01 |
| | 49 | | | | |
| **Member of a sports club in school (yes)** | 75 | 2.64 | 1.45 | 5.03 | < .01 |
| *No* | 55 | | | | |
| Sleep (N = 389) | | | | | |
| **Age (years)** | | 0.59 | 0.43 | 0.81 | < .01 |
| **How many breaks above >30 minutes do pupils get (one)** | 21 | | | | |
| *Two* | 25 | 1.13 | 0.62 | 2.09 | 0.69 |
| *Three or more* | 8 | 0.32 | 0.14 | 0.74 | 0.01 |
| **PE Sessions per week** | | 1.57 | 1.16 | 2.11 | < .01 |
| PA + Screen (n = 470) | | | | | |
| **Sex (female)** | 26 | | | | |
| *Male* | 36 | 1.64 | 1.10 | 2.47 | 0.02 |
| **Lunch break (Sat around)** | 25 | | | | |
| *Ran around* | 34 | 1.55 | 0.97 | 2.48 | 0.07 |
| *Stood around* | 47 | 2.76 | 1.06 | 7.15 | 0.04 |
| *Walked around* | 33 | 1.66 | 0.88 | 3.13 | 0.12 |
| PA + Sleep (N = 467) | | | | | |
| **County (Nairobi)*** | 5 | 0.27 | 0.07 | 0.95 | 0.04 |
| *Kitui* | 9 | | | | |
| **Sex (male)** | 9 | 2.18 | 0.97 | 4.93 | 0.06 |
| *female* | 5 | | | | |
| **Age (years)** | | 0.67 | 0.40 | 1.11 | 0.10 |
| **Can you ride a bicycle (yes)** | 8 | 4.42 | 0.95 | 20.51 | 0.06 |
| *No* | 2 | | | | |
| **Are you a member of school club (brownies guides or scout)** | | 2.36 | 1.03 | 5.41 | 0.04 |
| **Weight status (Healthy weight)** | 9 | | | | |
| *Below a healthy weight* | 4 | 0.33 | 0.11 | 0.97 | 0.04 |
| *Above a healthy weight* | 4 | 0.59 | 0.16 | 2.21 | 0.44 |

*(Continued)*

**Table 2.**  (Continued)

| | %[a] | OR | 95% CI | | p |
|---|---|---|---|---|---|
| **Sleep + Screen (n = 392)** | | | | | |
| **Sex (male)** | 0.5 | 0.12 | 0.02 | 0.96 | 0.05 |
| *Female* | 6 | | | | |
| **How many breaks above >30 minutes do pupils get (one** | 8 | | | | |
| *Two* | 1 | 0.16 | 0.03 | 0.80 | 0.03 |
| *Three or more* | 1 | 0.11 | 0.01 | 0.87 | 0.04 |
| **PA + Screen + Sleep (n = 469)** | | | | | |
| **Age (years)** | | 0.55 | 0.35 | 0.87 | 0.01 |
| **County (Nairobi)*** | 5 | 0.49 | 0.22 | 1.13 | 0.09 |
| *Kitui* | 10 | | | | |
| **Can you swim? (yes)** | 9 | 3.27 | 1.09 | 9.83 | 0.04 |
| *No* | 3 | | | | |
| **Screen time before school (no)** | 11 | 4.40 | 1.81 | 10.68 | 0.00 |
| *Yes* | 3 | | | | |

PE = Physical Education.

*All models containing county were additionally adjusted for age and weight status.

[a] Preparation of the reference category meeting the guideline.

CI = 0.14–0.74, p = 0.01). The odds of meeting sleep guidelines also decreased significantly with age (OR = 0.59, 95% CI = 0.43–0.81, p = 0.01).

## Discussion

The aim of this study was to investigate the proportion of children complying with the 24-h movement guidelines in urban and rural Kenya and to identify correlates of meeting these guidelines. Overall, compliance with the combined movement guidelines of PA, sedentary behaviour and sleep was low (8%). Indeed, more children met none of the guidelines (11%) than met all three. However, compliance with the combined movement guidelines was higher among children from rural (11%) compared to urban areas (6%). To the best of our knowledge, this is the first study in Kenya and only the second in SSA to compare compliance to the 24-h movement guidelines between children from rural and urban areas, using accelerometer measured movement behaviours.

Compliance with the combined movement guidelines among children from urban areas was consistent with the ISCOLE study (6.5%) [9]. However, the present study extends the existing literature by reporting a disparity between children from urban and rural areas, supporting similar findings from Mozambique [31]. Collectively, these studies demonstrate that in a region with a burgeoning urban population, there is an urgent need to improve movement behaviours in urban settings and/or slow migration from urban to rural areas. Nevertheless, in SSA the majority of the population continue to live in a rural setting. Therefore, these findings also emphasise the importance of including children from both urban and rural areas when the intention of research is to in part influence on government policy and strategy.

Kenya like many countries in SSA has experienced rapid urbanization in recent years [12]. Simultaneously, the burden of disease is shifting from communicable disease and undernutrition to NCD's such as cardiovascular disease and type 2 diabetes [32]. This epidemiological transition is underpinned by a change in dietary habits and habitual PA that underlie the distinct socioeconomic and environmental differences between rural and urban areas and leads

to an increased prevalence of overweight and obesity. Our finding that a significantly greater proportion of rural compared to urban children met the individual sleep and PA guidelines is largely consistent with previous studies of Kenyan children [13], adolescents [33], and from SSA more broadly [34] which show that rural children are more physically active and spend less time in sedentary behaviours than their urban counterparts. These differences have been attributed to more frequent engagement in outdoor chores such as gardening, herding livestock, fetching water, and collecting wood among rural children. In contrast, urban children report fewer active chores and more sedentary activities such as studying, watching television, and listening to the radio [33].

Although urban children would like to engage in active play, many reside in densely populated neighbourhoods characterised with high rise buildings with limited recreational space for play except on the streets which are congested with motor vehicle and human traffic [35, 36]. Many of these settings also have high crime rates and risks, forcing parents to keep their children indoors and engaged in less active pursuits. In agreement with this assertion, the present study demonstrated that children from urban areas were more likely to report using a device with a screen and doing homework before school. However, there was no difference in the prevalence of meeting ST guidelines between children from urban and rural areas which may suggest greater exposure to devices with screens and therefore opportunities for ST in rural areas. Notably, alongside high rates of rural to urban migration there is an increase of urbanization of the rural areas. The quest to quickly 'catch up' with the urban areas is linked to a sign of affluence and prestige. With these changes, the rural child's lifestyle may soon be similar to their urban counterparts, unless interventions and policies are developed to mitigate the crisis.

This study also identified several correlates associated with meeting the combined and individual movement guidelines. The odds of meeting the combined movement guidelines reduced with age, were significantly greater among those who could swim and among those who did not use a device with a screen prior to attending school. Importantly, urban and rural differences in the prevalence of meeting both the combined movement guidelines and the individual sleep and PA guidelines were not significant in the logistic regression models. This indicates that differences were largely explained by the observed correlates including sex, weight status and school level influences. Indeed, a novel finding of the present study was that adherence to sleep and PA guidelines was associated with features of the school day. Specifically, the number of PE classes provided per week was associated with higher odds of meeting the individual PA and sleep guidelines. Moreover, the odds of meeting the PA guidelines were higher among children who spent their lunch break walking or running around, compared to those who sat around but were lower among children from schools that provided transport for extracurricular activities. These findings, highlight the influence that schools have on movement behaviours. However, they also identify a clear contrast between urban and rural schools.

In Kenya, the frequency and duration of PE classes is dictated by government policy. However, previous studies have raised concerns over the compliance with the government policy and adherence to PE lessons in the timetable [37, 38]. It is suggested that PE is perceived to have inferior status relative to other disciplines and is often marginalised in favour of examinable subjects [38]. A lack of essential facilities, equipment and trained PE teachers have also been shown to hinder the implementation of the PE lessons [37]. PA has also been used as punishment which may have a detrimental impact on PA in the long-term by developing negative perceptions of sports and PA [39].

The PE syllabus strives to develop physically literate individuals who have the knowledge, skills, and confidence to enjoy activity across their life course [40]. Among schools that deliver

PE as scheduled, the amount of time dedicated to being active is unknown. Indeed, previous studies have shown that as little as 11% of PE classes were spent in MVPA [41]. Further research is needed to understand how much time is spent on PA and at what intensities. Nonetheless, the present study demonstrates that PE is an important correlate of meting PA guidelines and that PE classes were both shorter and less frequent in urban compared to rural schools. These findings support the urgent need to evaluate the delivery of PE in urban schools.

While school time allocated to PE is limited, recess is scheduled for more periods each day, making it an important opportunity for PA during the school day. In the present study, children who spent their lunch breaks walking or running had higher odds of meeting PA guidelines than those who spent it sitting. However, children from urban schools were less likely to be active during recess than their rural counter parts. One potential reason for the differences is lack of space in the school grounds and the very large pupil populations. A lack of space has been reported as a barrier to delivering PE in private schools with many using facilities outside the school grounds [37]. It is likely that a lack of space would also restrict opportunities for activity during recess as well as school sports programmes and PE classes.

After school clubs provide opportunities for self-directed PA or participation in sport and have been associated with greater PA [42]. Despite high engagement with after school clubs in the present study, their attendance did not increase the odds of meeting PA guidelines. However, being a member of a school sports club was associated with greater odds of achieving the screen guidelines. The reason behind this is unclear, however, it is possible that through engagement with clubs, children displace time that could otherwise be spent in ST. embraces.

Taken together, these findings support a whole school approach to PA premotions as has been demonstrated by the Crating Active Schools and CSPAS in the United Kingdom and United States respectively [43, 44]. The whole school approach strives to create a physical and social environment that supports PA and increase opportunities for PA before, during and after school. This model extends the responsibility for PA premotion, beyond PE teachers, to all stakeholders, staff, students, parents, and community.

Swimming and cycling are both foundational movement competencies [45] and promoters of healthy active lifestyle; whilst being able to swim is important to reduce the risk of drowning [46], having the ability to cycle can improve safety on the road [47]. Within the current study children reporting that they could swim were 3.27 times more likely to meet the combined movement guidelines. Further, the odds of meeting the PA and sleep guidelines were greater among children who could ride a bike (OR = 4.42, 95% CI = 0.95–20.51, p = 0.06). When considering swimming and cycling as variables associated with meeting PA guidelines, a recent study by Richards et al. (2021) [48] concluded that the ability to swim and cycle were associated with all components of fitness in children. This further strengthens the link between fitness, PA and motor competence proposed in Stodden's (2008) [49] conceptual model which advocates for the development of foundational movement competencies as a driver for engagement in PA.

Limitations of this study include its cross-sectional design which precludes inferences about directionality or causation. Furthermore, the correlates identified are limited to the available variables and as such, potential confounding effects of unmeasured variables cannot be dismissed. This study also relied on self-reported data obtained by instruments whose validity have not been assessed in this population. Nonetheless, important strengths of this study include the relatively large sample size, the recruitment of children in rural, urban areas and the use of device-based measures of movement behaviours. Specifically, wrist-worn accelerometers increase wear time compliance and provide more accurate estimates of sleep duration compared with hip-worn devices [50]. Finally, the inclusion of school level correlates provides

insight into environmental and policy level factor influence movement behaviours while the multilevel analyses accounted for the hierarchical structure of these data.

## Conclusions

The prevalence of meeting movement guidelines among Kenyan children is low and of greatest concern in urban areas. Efforts to improve movement behaviours are urgently needed. The present study provides support for the school setting as an important focus for future intervention to improve compliance with movement guidelines.

## Supporting information

**S1 Table. Potential correlates of meeting movement guidelines.**
(DOCX)

**S2 Table. Univariate analysis of correlates of meeting the combined movement guidelines.**
(DOCX)

**S1 File. Inclusivity in global research.**
(DOCX)

## Acknowledgments

We thank the participants and school staff for their voluntary participation in this study. Without their contribution this research would not be possible.

## Author Contributions

**Conceptualization:** Nils Swindell, George Owino, Sinead Brophy, Huw Summers, Stuart J. Fairclough, Vincent Onywera, Gareth Stratton.

**Data curation:** Nils Swindell, Lucy-Joy Wachira, Victor Okoth, Stanley Kagunda, George Owino.

**Formal analysis:** Nils Swindell.

**Funding acquisition:** Nils Swindell, Lucy-Joy Wachira, George Owino, Sinead Brophy, Huw Summers, Vincent Onywera, Gareth Stratton.

**Investigation:** Nils Swindell, Lucy-Joy Wachira, Victor Okoth, Stanley Kagunda, Sophie Ochola, Vincent Onywera.

**Methodology:** Nils Swindell, Lucy-Joy Wachira, Victor Okoth, Sophie Ochola, Sinead Brophy, Huw Summers, Vincent Onywera, Gareth Stratton.

**Project administration:** Nils Swindell, Lucy-Joy Wachira, Victor Okoth, Stanley Kagunda, Sophie Ochola, Vincent Onywera.

**Supervision:** Nils Swindell, Lucy-Joy Wachira, George Owino, Sophie Ochola, Sinead Brophy, Stuart J. Fairclough, Vincent Onywera, Gareth Stratton.

**Validation:** Amie Richards.

**Writing – original draft:** Nils Swindell.

**Writing – review & editing:** Lucy-Joy Wachira, Victor Okoth, Stanley Kagunda, George Owino, Sophie Ochola, Sinead Brophy, Huw Summers, Amie Richards, Stuart J. Fairclough, Vincent Onywera, Gareth Stratton.

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
