## [Decision Letter · Decision Letter 0]

24 Oct 2022

PONE-D-22-24523Prevalence and correlates of compliance with 24-h movement guidelines among children from urban and rural Kenya – The Kenya-LINX ProjectPLOS ONE

Dear Dr. Swindell,

Thank you for submitting your manuscript to PLOS ONE. After careful consideration, we feel that it has merit but does not fully meet PLOS ONE’s publication criteria as it currently stands. Therefore, we invite you to submit a revised version of the manuscript that addresses the points raised during the review process.

Dear Author,

This manuscript still requires a few minor amendments to be made, please make the necessary changes.

The decision of this manuscript is justified based on PLOS ONE’s publication criteria and not on its novelty or perceived impact.

We look forward to receiving your revised manuscript.

Kind regards,

Zulkarnain Jaafar

Academic Editor

PLOS ONE

Journal Requirements:

Reviewers' comments:

Reviewer's Responses to Questions

**Comments to the Author**

1. Is the manuscript technically sound, and do the data support the conclusions?

Reviewer #1: Yes

Reviewer #2: Yes

2. Has the statistical analysis been performed appropriately and rigorously? 

Reviewer #1: Yes

Reviewer #2: Yes

3. Have the authors made all data underlying the findings in their manuscript fully available?

Reviewer #1: No

Reviewer #2: No

4. Is the manuscript presented in an intelligible fashion and written in standard English?

Reviewer #1: Yes

Reviewer #2: Yes

5. Review Comments to the Author

Reviewer #1: As the title is the second published work so far in Sub-Saharan countries, it is well done. However, the format should follow PLOS ONE guideline. Hence, I suggest the authors to include limitation of the study, abbreviation and authors contribution content.

Line 262, table 1, row 5: avoid such spacing which will affect the journal and the paper reputation

Spacing between headings are not uniform and blank page should be avoided.

New paragraph should be double spaced from the other.

Corresponding author address should be full

Reviewer #2: This study examined compliance to the 24-hourmovement guidelines and their correlates among children from urban and rural Kenya. I would like to congratulate Authors for this interesting study. In general, the present study is worth publishing and adds valuable information. Nevertheless, I consider there are some changes that should be done.

Page 5, line 106: Please, explain an abbreviation of PE by providing its full form.

Page 5, line 120: Please, give more detailed information about the participants – ho did Authors recruit them (probability or convenience sampling)?

Page 19, line 403 Several studies for me means more than two. Please add some references here.

6. PLOS authors have the option to publish the peer review history of their article (what does this mean?). If published, this will include your full peer review and any attached files.

Reviewer #1: **Yes: **Eyob Girma Abera

Reviewer #2: No

---

## [Author Response · Author response to Decision Letter 0]

20 Nov 2022

Dear Zulkarnain Jaafar, 

On behalf of my co-authors, I would like to thank you for the opportunity to revise and resubmit our manuscript entitled Prevalence and correlates of compliance with 24-h movement guidelines among children from urban and rural Kenya – The Kenya-LINX Project. We have carefully considered the reviewers comments and adjusted the manuscript accordingly. Below, we have addressed each comment and provided a response. Corresponding changes are highlighted in the manuscript using track changes. 

Editor comments 

Response: Thank you for the guidance, the manuscript has been adjusted to follow the templates provided. 

Response: Questionnaire has been completed and attached. 

3. In your Data Availability statement, you have not specified where the minimal data set underlying the results described in your manuscript can be found. PLOS defines a study's minimal data set as the underlying data used to reach the conclusions drawn in the manuscript and any additional data required to replicate the reported study findings in their entirety. All PLOS journals require that the minimal data set be made fully available. For more information about our data policy, please see http://journals.plos.org/plosone/s/data-availability. Upon re-submitting your revised manuscript, please upload your study’s minimal underlying data set as either Supporting Information files or to a stable, public repository and include the relevant URLs, DOIs, or accession numbers within your revised cover letter. For a list of acceptable repositories, please see http://journals.plos.org/plosone/s/data-availability#loc-recommended-repositories. Any potentially identifying patient information must be fully anonymized. Important: If there are ethical or legal restrictions to sharing your data publicly, please explain these restrictions in detail. Please see our guidelines for more information on what we consider unacceptable restrictions to publicly sharing data: http://journals.plos.org/plosone/s/data-availability#loc-unacceptable-data-access-restrictions. Note that it is not acceptable for the authors to be the sole named individuals responsible for ensuring data access. We will update your Data Availability statement to reflect the information you provide in your cover letter.

Response: We are unable to make the data available on ethical grounds as ethical approvals and subsequently consent was obtained on the basis that data would only be shared with the research team. Specifically, the parent consent form contained the following statement “The information you or your child provides will be kept private and the only people who will have access to the information are members of the research team”.

Revised data availability statement:

Ethical approval for this study was granted by the Swansea University College of Engineering Research Ethics Committee and The Kenyatta University Ethical Review Committee, provided that participants’ data was only accessible by the research team. Data from this research study contains information at the individual and school level which when combined could be used to identify individuals. Participants of this study did not consent to having their data publicly available. 

Requests to access the data may be directed to the Swansea University College of Engineering Research Ethics Committee coe-researchethics@swansea.ac.uk and the Ethics and scientific review committee at Kenyatta University chairman.kuerc@ku.ac.ke, APPLICATION NUMBER PKU2106/11254

Response: Thank you for pointing this out, captions have been included on page 29 lines 670-673.

Response: Thank you for this comment. We have checked each reference to ensure they are still available and relevant. To our knowledge none of the references have been retracted, however, the following changes have been made. 

Page 24, 

Reference 15, the name of the journal has been included in full. 

Reference 16, the title has been edited to remove a typo. 

Reference 30 capitalisation of title and the Doi have been corrected 

Reference 50 – Correction made to the title 

Reviewers' comments:

Reviewer's Responses to Questions

Comments to the Author

1. Is the manuscript technically sound, and do the data support the conclusions?

Reviewer #1: Yes

Reviewer #2: Yes

2. Has the statistical analysis been performed appropriately and rigorously? 

Reviewer #1: Yes

Reviewer #2: Yes

3. Have the authors made all data underlying the findings in their manuscript fully available?

Reviewer #1: No

Reviewer #2: No

Response: Thank you for taking the time to review our manuscipt and for the positive commenets. 

As descibed above, we are not able to make the data available on ethical grounds as ethical approvals and subsequently consent was obtained on the basis that data would only be shared with the research team. The data availability statement has been updated to reflect the status of availability and offer contact information for the two university ethical committees. 

4. Is the manuscript presented in an intelligible fashion and written in standard English?

Reviewer #1: Yes

Reviewer #2: Yes

5. Review Comments to the Author 

Reviewer #1: As the title is the second published work so far in Sub-Saharan countries, it is well done. However, the format should follow PLOS ONE guideline. Hence, I suggest the authors to include limitation of the study, abbreviation, and authors contribution content.

Response: Thank you for these suggestions. Study limitations can be found on page 21 lines 460 to 476. Author contributions have been added on page 29 line 671. Abbreviation have now all been described in full on first use (edit made line 110).

• Line 262, table 1, row 5: avoid such spacing which will affect the journal and the paper reputation

• Thank you for spotting this, spacing has not been adjusted. 

Spacing between headings are not uniform and blank page should be avoided.

New paragraph should be double spaced from the other.

Response: Thank you for the comments, spacings have been adjusted throughout the document 

• Corresponding author address should be full 

Response: Thank you for this comment, the details of the corresponding author have been adjusted to conform with the author guidelines (line 22) https://journals.plos.org/plosone/s/file?id=ba62/PLOSOne_formatting_sample_title_authors_affiliations.pdf. However, these guidelines did not recommend including address in full. If the address is required elsewhere in the submission, kindly advise. 

Reviewer #2: This study examined compliance to the 24-hourmovement guidelines and their correlates among children from urban and rural Kenya. I would like to congratulate Authors for this interesting study. In general, the present study is worth publishing and adds valuable information. Nevertheless, I consider there are some changes that should be done.

Response: Thank you for taking the time to review our manuscript and for the positive comments. 

• Page 5, line 106: Please, explain an abbreviation of PE by providing its full form. Response: Thank you for spotting this, this has been adjusted on line 110. 

• Page 5, line 120: Please, give more detailed information about the participants – how did Authors recruit them (probability or convenience sampling)?

Response: Thank you for this comment. The sampling framework used stratified random cluster sampling. Schools were randomly selected after being stratified by county (Nairobi/Kitui), type of school (private/public) and sub-county. As for the children, we used a full class participation approach of children with the eligible age and that had signed parental consent. We have included a line to describe this process (page 6 lines 130-132). 

• Page 19, line 403 Several studies for me means more than two. Please add some references here.

Response: We agree with this comment and have edited the manuscript accordingly (page19, line 411)

---

## [Decision Letter · Decision Letter 1]

14 Dec 2022

Prevalence and correlates of compliance with 24-h movement guidelines among children from urban and rural Kenya – The Kenya-LINX Project

PONE-D-22-24523R1

Dear Dr. Swindell,

We’re pleased to inform you that your manuscript has been judged scientifically suitable for publication and will be formally accepted for publication once it meets all outstanding technical requirements.

Kind regards,

Zulkarnain Jaafar

Academic Editor

PLOS ONE

Additional Editor Comments (optional):

Reviewers' comments:

Reviewer's Responses to Questions

**Comments to the Author**

1. If the authors have adequately addressed your comments raised in a previous round of review and you feel that this manuscript is now acceptable for publication, you may indicate that here to bypass the “Comments to the Author” section, enter your conflict of interest statement in the “Confidential to Editor” section, and submit your "Accept" recommendation.

Reviewer #1: All comments have been addressed

Reviewer #2: All comments have been addressed

2. Is the manuscript technically sound, and do the data support the conclusions?

Reviewer #1: Yes

Reviewer #2: Yes

3. Has the statistical analysis been performed appropriately and rigorously? 

Reviewer #1: Yes

Reviewer #2: Yes

4. Have the authors made all data underlying the findings in their manuscript fully available?

Reviewer #1: Yes

Reviewer #2: No

5. Is the manuscript presented in an intelligible fashion and written in standard English?

Reviewer #1: Yes

Reviewer #2: Yes

6. Review Comments to the Author

Reviewer #1: I have seen all the documents and the requested change has made. I suggest the paper to be published.

Reviewer #2: (No Response)

7. PLOS authors have the option to publish the peer review history of their article (what does this mean?). If published, this will include your full peer review and any attached files.

Reviewer #1: **Yes: **Eyob Girma Abera

Reviewer #2: No

---

## [Editor Report · Acceptance letter]

21 Dec 2022

PONE-D-22-24523R1 

Prevalence and correlates of compliance with 24-h movement guidelines among children from urban and rural Kenya – The Kenya-LINX Project 

Dear Dr. Swindell:

I'm pleased to inform you that your manuscript has been deemed suitable for publication in PLOS ONE. Congratulations! Your manuscript is now with our production department. 

Kind regards, 

on behalf of

Dr. Zulkarnain Jaafar 

Academic Editor

PLOS ONE